# Development of a Zeolite A/LDH Composite for Simultaneous Cation and Anion Removal

**DOI:** 10.3390/ma12040661

**Published:** 2019-02-22

**Authors:** Breno Gustavo Porfírio Bezerra, Lindiane Bieseki, Djalma Ribeiro da Silva, Sibele Berenice Castellã Pergher

**Affiliations:** 1Posgraduate Program in Chemistry, Chemistry Institut, Federal University of Rio Grande do Norte. Av Senador Salgado Filho, 3000. CEP 59078-970 Natal/RN, Brazil; brenogpb@gmail.com (B.G.P.B.); djalmarib@gmail.com (D.R.d.S.); 2Molecular Sieves Laboratory, Chemistry Institut. Av Senador Salgado Filho, 3000. CEP 59078-970 Natal/RN, Brazil; lindiane.bieseki@gmail.com

**Keywords:** water produced, adsorbent materials, composite

## Abstract

Wastewater from the oil industry is a major problem for aqueous environments due to its complexity and estimated volume of approximately 250 million barrels per day. The combination of these petroleum pollutants creates risks to human health, and their removal from the environment is considered a major problem in the world today. Thus, this work has the objective of studying the treatment of this type of effluent through the adsorption method using the following exchange materials: cationic, anionic, their combination by a sequential method, and a composite material. Zeolite A, a layered double hydroxide (LDH), and the new composite material formed by zeolite A and LDH structures were synthesized for this study. All were used for the simultaneous treatment of cations and anions in a complex sample such as water produced from petroleum production. The composite demonstrated an excellent ability to simultaneously remove cations and anions. The results obtained after the different treatment modes of the effluent using different materials varied from 85% to 100% for the removal of cations and from 56% to 99.7% for the removal of anions.

## 1. Introduction

The regular operations of the oil and gas industry are characterized by a large amount of water injected to facilitate the recovery of oil. This water is brought to the surface along with hydrocarbons (oil and gas), salt and other solutes, and it is commonly known as “produced water” [1]. The liquid waste stream produced by the oil industry is estimated to be approximately 250 million barrels per day, and the water to oil ratio is at least 3:1. The complex composition of the produced water (PW) is variable and depends on the natural geological characteristics of the location. Its properties can vary, as it contains several soluble mineral ions and is often of an acidic nature [2]. Some constituents of concern in PW are salts (expressed as salinity), total dissolved solids, oil and grease content, natural inorganic and organic compounds (e.g., chemicals that cause hardness and scaling such as calcium, magnesium, sulphates and barium) and the chemical additives used in drilling, fracturing and well operation, which may have some toxic properties (e.g., biocides and corrosion inhibitors) [3]. The inorganic minerals, which are present as dissolved salts, dissolved in the PW are strongly related to the geochemical characteristics of the well, and they are the cations Na^+^, K^+^, Ca^2+^, Mg^2+^, Ba^2+^, Sr^2+^, and Fe^2+^ as well as the anions Cl^−^, SO_4_^2−^, CO_3_^2−^, and HCO_3_^−^, which affect the PW chemistry in terms of the buffering capacity, salinity and potential for scaling [4]. The toxic metals commonly found in PW are cadmium, chromium, copper, lead, mercury, nickel, silver and zinc, and they are mainly of natural origin [4]. 

Wastewater from the petroleum industry is a major problem for aqueous environments. The combination of these petroleum pollutants creates risks to human health [5]. Their removal from the environment is considered a major problem in today’s world [6]. The reuse or recycling of water has become mandatory, especially in countries with water concerns. Many countries have implemented more stringent regulations for the permitted limits of oil and gas (O and G) that can be discharged into the produced water, ranging from 10 mg L^−1^, according to China’s Ministry of Environment, up to the maximum limit of 42 mg L^−1^, regulated by the United States Environmental Protection Agency (USEPA) [7]. In Brazil, in environmental legislation, the maximum value allowed is up to a 20 mg L^−1^ value defined through CONAMA-National Environment Council, with resolutions no. 357 (CONAMA 357/2005) [8] and no. 430 (CONAMA 430/2011) [9], which establishes the conditions and standards for the discharge of effluents by determining the maximum limits of concentrations of the parameters. New regulations have promoted the development of ecological and economic disposal methods [10]. This can be seen as an opportunity to treat the water produced and provide a viable source of water, which is beneficial in many applications, for which the quality of drinking water is not necessary while avoiding serious environmental damage. Thus, this work has the objective of PW treatment by adsorption method using a composite material that can be capable of removing simultaneously cations and anions. 

It is well known that zeolites can remove cations and layered double hydroxides (LDHs) can remove anions. So these materials can be used for PW treatment. In this study, a composite made by zeolite A and LDH was synthesized and used for PW treatment. Previous studies have performed the synthesis of a zeolite A and LDH composite. Yamada et al. [11] coated a sample of zeolite A with LDH through the dripping of a mixture of solutions of magnesium chloride and aluminium chloride but did not study its application. Another study was carried out in order to synthesize a zeolite A and LDH (Mg/Fe) composite using the acid residue of the copper and kaolin processing process for the combined synthesis of zeolite A and LDH. The synthesis occurred in the following two steps: synthesis of LDH by co-precipitation and then the addition of metakaolin for the synthesis of zeolite A [12]. 

In this study, zeolite A, LDH, and a new composite material formed by zeolite A and LDH (Mg/Al) structures were synthesized separately. The synthesis process of the composite was carried out at room temperature in a shorter synthesis time than in the literature [11,12], with the excellent formation of the crystals. In addition to the new synthesis processes, this study used for the first time the synthesized materials for the simultaneous removal of cations and anions in wastewater produced from petroleum, which is considered a highly complex removal process. Zeolite A is characterized by the presence of molecular-sized pores and cavities, which are occupied, due to charge compensation, by cations and water molecules. These ions are not covalently bound to the structure, which makes the zeolites excellent cation exchangers [13,14,15]. LDH is an anionic clay with positive layers of di- and trivalent metal and ions and anions in the interlayer region present to neutralize this charge. The freedom of these intercalary anions causes the LDH to have the capacity of an anion exchanger [16,17]. Thus, the composite material is ideal for the reutilization of produced water by the simultaneous removal of the cations and anions present.

## 2. Materials and Methods

### 2.1. Synthesis and Characterization of the Materials

For the preparation of the composite based on zeolite A and LDH, two synthetic methodologies were used sequentially. The first methodology was based on the synthesis of IZA for zeolite A [18], and the second methodology was based on the procedure proposed by Climent and co-workers [19], with changes including seeking an LDH with an Mg/Al = 3 ratio. A sample of pure zeolite A and another of LDH were also prepared for comparison with the synthesized composite. 

The preparation of the composite consists of two fundamental steps. In the first step, a gel was prepared with the following composition: 0.098 mol SiO_2_, 0.049 mol Al_2_O_3_, 0.157 mol Na_2_O and 6.285 mol H_2_O. To prepare the gel, 1.266 g of NaOH (Sigma Aldrich) was added to 100 g of Milli-Q H_2_O, and after complete dissolution, the solution is divided into two equal volume fractions, called V1 and V2. In the first fraction V1, 9.100 g of sodium aluminate (50%–56% Al_2_O_3_ 40%–45% Na_2_O) was added until dissolution. In the second fraction V2, 13.225 g of deionized H_2_O plus 6.000 g of NaOH and 5.900 g of SiO_2_ (Aerosil 200 Degussa) were added. After the homogenization of both fractions, fraction V1 was rapidly added to fraction V2. After stirring the system for approximately 30 min, the prepared gel was placed in Teflon autoclaves, where it was kept in a hot block for 4 h at 100 °C.

During the crystallization of the gel, a solution of Mg(NO_3_)_2_·6H_2_O at 0.225 mol·L^−1^ was prepared with Al(NO_3_)_2_·9H_2_O at 0.075 mol·L^−1^ in a final volume of 200 mL (solution A). A 0.4 mol·L^−1^ NaOH solution was also prepared with 0.2 mol L^−1^ Na_2_CO_3_ to a final volume of 400 mL (solution B). 

For the second step, the gel obtained in the first step was transferred to a stirred glass beaker, in which solution (A) containing the magnesium and aluminium nitrates and solution (B) containing the sodium carbonate and sodium hydroxide were dripped in simultaneously. The dripping occurred slowly under agitation on the synthesis gel at a constant temperature of 60 °C. At the end of the addition of solutions A and B, the mixture was stirred for 3 h at 60 °C for ageing. After this time, the solid obtained was separated by filtration and washed with distilled water until pH = 7, and it was finally dried at room temperature. The material produced was denominated ZAHD composite and was also calcined in air with a 3 °C/min heating ramp at 400 °C for 3 h. The composite before and after being calcined was then characterized.

The prepared composite material was characterized by differential thermal and thermogravimetric analysis (DTG, DTA) under a dynamic nitrogen atmosphere with a flow rate of 20 mL min^−1^ at a heating rate of 10 °C·min^−1^ from 30 and 800 °C using a mass of approximately 10 mg deposited in a platinum sample port. The DTG curves were obtained by calculating the first derivative of the TG curves.

The samples synthesized in this work were characterized by a Bruker D2 Phaser diffractometer (Billerica, MA, USA) using Cu radiation (λ = 1.54 Å) with a step size of 0.02°, a current of 10 mA, and a voltage of 30 kV. A Lynxeye detector (192 channels), divergent 0.6 mm slit, 0.1 s time, and 1 mm anti-air scattering screen was used.

In the Chemical analysis, the samples were characterized in a Bruker S2 Ranger apparatus (Billerica, MA, USA), using P power radiation, a Ag power of 50 W, a maximum voltage of 50 kV, a maximum current of 2 mA, and a XFlash® Silicon Drift Detector (Billerica, MA, USA), with the results were normalized to 100%.

The samples were analysed using a ZEISS brand electronic scanning electron microscope (Oberkochen, German), an Auriga model FEG (field emission gun) type emitter, with a voltage of 20 kV, a chemical analysis detector using energy dispersion spectroscopy (EDS, Bruker, Billerica, MA< USA) and a XFlash® 410-M detector (Bruker, Billerica, MA< USA).

The specific area analysis and N_2_ adsorption isotherm were obtained by the physical adsorption of nitrogen on the material by the Brunanuer-Emmett-Teller method (BET). This method is based on the determination of the volume of N_2_ adsorbed at various relative pressures at the temperature of liquid nitrogen, at pressures of up to 2 atm and at relative pressures (p/p_0_) of less than 0.3 atm. For the performance of this test, a specific area metre, a Micromeritcs brand ASAP 2020 (Altalnta, GA, USA), was used. The data were taken from the nitrogen sorption isotherms at 77 K to determine the specific surface area (S_BET_ and Gurvich rule for the total pore volume, VTP).

### 2.2. Treatment of Water Produced from Petroleum

The treatment of the water produced from oil (from the RN CE Operation Unit) was performed using 50 mL of the solution in contact with a mass of 0.30 g of each material. The samples were placed in 150 mL Erlenmeyer flasks and subjected to a stirring orbital shaker table operating at 200 rpm for periods equal to 4 h. At the end of this time, the samples were filtered and analysed by IC and ICP-OES. 

The first treatment was with LDH and then with zeolite A, while in the second treatment, the order was reversed with zeolite A being used before LDH. In the last treatment, the composite was used.

Cation and anion concentrations were determined by liquid chromatography (IC) and by inductively coupled plasma optical emission spectrometry (ICP OES). The analyses were performed in triplicate.

The produced water sample was analysed by inductively coupled plasma optical emission spectrometry (ICP OES). The equipment used was an iCAP 6300 Duo model (Thermo Fisher Scientific, Massachusetts, USA) with axial and radial views and a simultaneous CID (charge injection device) Detector. Commercial argon with a purity of 99.996% (White Martins-Praxair) was used along with the following parameters: power source RF at 1150 W, nebulizer gas flow at 0.75 L min^−1^, auxiliary gas flow at 0.5 L min^−1^, and stabilization time of 15 s.

The sample of water produced was also analysed by ion chromatography (IC) using an ICS-2000 DIONEX ion chromatograph (Sunnyvale, CA, USA), with an in situ eluent generator, conductivity detector and electrochemical suppression as well as an AS40 DIONEX autosampler. The analytical column and guard column used were an IonPAC AS19 2 × 250 mm and an IonPAC AG19 2 × 50 mm, respectively, both from DIONEX.

## 3. Results

### 3.1. Synthesis and Characterization

Figure 1 shows the diffractograms and the structure scheme of the following materials: LDH, zeolite A and zeolite A/LDH composite (ZAHD).

Only the LDH phase is present in the diffractogram of Figure 1, where the characteristic reflections are observed at 2θ = 11.34°, 22.88°, 34.35°, 38.73° and 45.73°, with the value of d(003) = 7.8 Å being due to the presence of carbonate between the lamellae [20,21]. The sample of zeolite A (Figure 1) is pure zeolite A and presents all of the characteristic reflections of this phase, with it then being used as a standard [22]. The diffractogram of the composite (Figure 1) shows characteristic reflections of the LDH and zeolite A compounds, corresponding to the new composite (ZAHD).

Figure 1 shows a scheme of the materials structures and the proposed structure of the composite, that is zeolite A crystals covered by LDH layers (see also SEM results in Figure 3).

A thermogravimetric analysis of the composite was performed with the objective of observing the changes in relation to the loss of mass occurring upon heating. The TG/DTG curves and mass loss percentages for each event are shown in Figure 2.

To confirm the formation of the composite, high resolution scanning electron microscopy (SEM) analyses were performed. LDH, zeolite A and ZAHD composite samples were analysed, and the images are presented in Figure 3.

The micrographs of zeolite A and LDH are in agreement with the literature according to Reference [19]. The zeolite A presents a morphology of cubic crystals with defined edges [23]. For the LDH, a morphology showing a series of hexagonally squamous particles is indicated. In the synthesis of the composite, from the enlargement, it can be observed that it has a well-defined morphology resembling cubic crystals, but its surface has a smooth homogeneous layer, leaving these crystals without a defined format but totally covered. 

Table 1 presents the FRX results for the three studied materials: zeolite A, LDH, and the composite (ZAHD).

The chemical composition for zeolite A is in agreement with that observed for this material in the literature [24]. 

The Mg/Al ratio for the LDH was 1.9 and that value was lower than expected (3.0) The Mg/Al ratio for the composite was 0.77, while the Si/Al ratio was equal to 0.83.

In the determinations of the specific surface areas, the nitrogen adsorption/desorption technique was used at 77 K, the isotherms for the synthesized materials are presented in Figure 4.

Figure 4 shows that the LDH isotherm is of type II, which is characteristic of non-porous materials. This occurs because the lamella dimensions are larger than the relative size of the nitrogen molecule [25,26]. The zeolite A isotherm shown in Figure 4 is classified as being type I, presenting a low specific area due to the size of its micropores [27]. In Figure 4, the composite isotherm is type IV, which is characteristic of porous and lamellar materials. Mathematical models can be applied to determine the specific surface area and total pore volume. Table 2 shows the results obtained.

The BET surface area value and micropore volume calculated for the composite are lower than those obtained for the pure LDH, and this is justified by the fact that a large amount of the composite is formed by zeolite A.

### 3.2. Treatment of a Sample of Water Produced from Petroleum

Knowing the properties of the LDH, zeolite A and the composite material, in this work the treatment of samples of petroleum produced water with different concentrations of cations and anions was carried out. Table 3 shows these results. The first treatments were using only Zeolite A and LDH; then two treatments performed sequentially form LDH/zeolite A, and the reverse order forms zeolite A/LDH, with the last sample used being the composite. Figure 5 shows the ICP-OES results in relation to the removal of cations for the different treatments employed.

The initial concentrations of Al, Cd, Cu, Mn, Ni, Pb and Zn ions without the presence or action of adsorbents were 3.913 mg L^−1^, 0.448 mg L^−1^, 0.385 mg L^−1^, 2.319 mg L^−1^, 0.482 mg L^−1^, 0.437 mg L^−1^ and 2.407 mg L^−1^. The removal percentages obtained in the treatment performed by the composite were 99.2%, 95%, 100%, 85%, 100%, 100% and 98.8% for Al, Cd, Cu, Mn, Ni, Pb and Zn, respectively.

Chloride, bromide, nitrite, nitrate, phosphate and sulphate concentrations present in water produced during treatment with the different treatments, as analysed by IC, are presented in Figure 6. 

The initial concentrations of chloride, bromide, nitrite, nitrate, phosphate and sulphate without the presence or action of adsorbents were 203.971 mg L^−1^, 18.656 mg L^−1^, 13.97 mg L^−1^, 60.757 mg L^−1^, 18.304 mg L^−1^ and 19.027 mg L^−1^. The removal rates obtained by the composite treatment were 85.7%, 57%, 56%, 77.4%, 99.7% and 70.1% for the chloride, bromide, nitrite, nitrate, phosphate and sulphate ions, respectively.

## 4. Discussion

Comparing the X-ray diffraction results of the LDH and zeolite A samples with the ZAHD composite sample, it is observed that the reflections present coincide with the two phases mentioned above. It is observed that the intensities of the reflections related to the zeolite A are less intense than the standard material, which may be related to the later stage of formation of the LDH phase, where an alkaline system is used. In the formation of the composite, the crystals of zeolite A are surrounded by the layered double hydroxide. This may also cause the reflections of the zeolite to be less intense. In contrast from the study conducted by Yamada et al. (2006) [11], the reflections of LDH can be clearly identified, especially reflections corresponding to the 003 and 006 planes.

In the thermogravimetric analysis, two mass losses with three events were observed, which can be observed in Figure 2. The first loss of mass occurred between 30–100 °C and was 5.07%, which was attributed to the removal of poorly adsorbed water due to the low temperature. The second mass loss occurring between 100–200 °C represented 15.41% of the total mass. The temperature at which this event occurs suggests that the loss is due to the removal of the interlamellar water due to the presence of LDH and the poorly adsorbed water in the zeolite structure. A third mass loss occurred at approximately 200–400 °C (where the maximum loss occurs), which is within the correct temperature range for dehydroxylation and decarbonation. So this temperature (400 °C) was used for calcination. This is the largest mass loss, accounting for 20.09% of the total mass lost [28]. It is common that the higher mass loss during the thermal decomposition of a present LDH results from the simultaneous dehydroxylation and removal of anhydrous interlayers of [29]. 

In the micrographs shown in Figure 3, it is observed that the LDH particles are inclined to strongly agglomerate together. The thickness of the squamous particles is very small, only a few nanometres, which has already been observed in other studies [11,12,30]. In our study, due to the composite synthetic process being carried out from the crystallized material but still dispersed in the aqueous synthetic matrix (mother water), a thicker layer was formed around the crystals of zeolite A.

The Si/Al and Mg/Al ratio data presented for the composite in Table 1 are lower than those obtained for zeolite A and LDH, respectively. This value is down because it is estimated using the amount of total aluminium, i.e., the amount present in zeolite A and HDL. The amount of NaO_2_ present is lower in the composite compared to pure zeolite A, which may indicate that a portion of the Na present in the zeolite structure may have been replaced by Mg. 

The formation of the composite is supported by the adsorption data, where we can observe that the specific surface area obtained was in between those of LDH and zeolite A (Table 2). The value of 113 m^2^·g^−1^ for LDH may be related to the synthetic temperature employed. Lower temperatures favour the formation of LDH-type materials with higher specific surface areas [31].

These materials, Zeolite A, LDH and composite; are materials with charge deficiency. So the mechanism that will occur in the adsorption process will be the exchanging of cations and anions. So, in the case of zeolites, the molar ratio Al/Si is important, because the Al will give a negative charge to the framework, so with more aluminium, more cationic exchange capacity the material will have. And in the case of LDH, the Al gives a positive charge to the framework, so with more Al, more anionic exchange capacity the material will have. 

In the produced water (PW) treated only with zeolite A (0.3 g for 4 h), approximately all of the cations are removed (~100%), and the anions were not removed. When the PW was treated only with LDH (3 g for 4 h), approximately 80% to 90% of anions were removed (the only sulphate was ~25%), and the cations were not removed (Table 3). Thus, to simultaneously remove cations and anions from the PW, it will be necessary to employ both materials, LDH and zeolite A. The result showed that the sequential treatments have the same results, independent of the order in which they were used (60 to 100% for anions and 80% to 100% for cations). In these treatments, 0.3 g of each material was used for 4 h, and thus, the total used was 0.6 g for 8 h. When the composite was used, it was 0.6 g for 4 h, and it demonstrated the same capacity for removing cations and anions as the sequential treatment. Thus, the composite material is efficient for removing cations and anions from PW simultaneously and can do so in only 4 h. Another advantage of the composite material is that this material has a morphology and particle sizes more homogeneous than the use of zeolite A and LDH separately, sequentially or mixed. This fact can help to avoid diffusion problems. 

## 5. Conclusions

In this study, the syntheses of zeolite A, layered double hydroxide (LDH), and a composite material based on zeolite A and layered double hydroxide, called composite (ZAHD), were successfully performed at room temperature in only 7 h.

The materials zeolite A, LDH and the composite (ZAHD) were used for the simultaneous treatment of cations and anions in a complex sample of water produced from petroleum, and it was observed from the results that all of the materials have very good adsorption capacities. The results obtained after the different treatment modes of the effluent in different materials varied from 80 to 100% for the removal of cations and from 60% to 100% for the removal of anions. These values are extremely satisfactory considering the complexity of the effluent produced from petroleum, and since the concentrations of cations and anions are in the range of mg L^−1^ to μg L^−1^, treatment is even more difficult.

## Figures and Tables

**Figure 1 materials-12-00661-f001:**
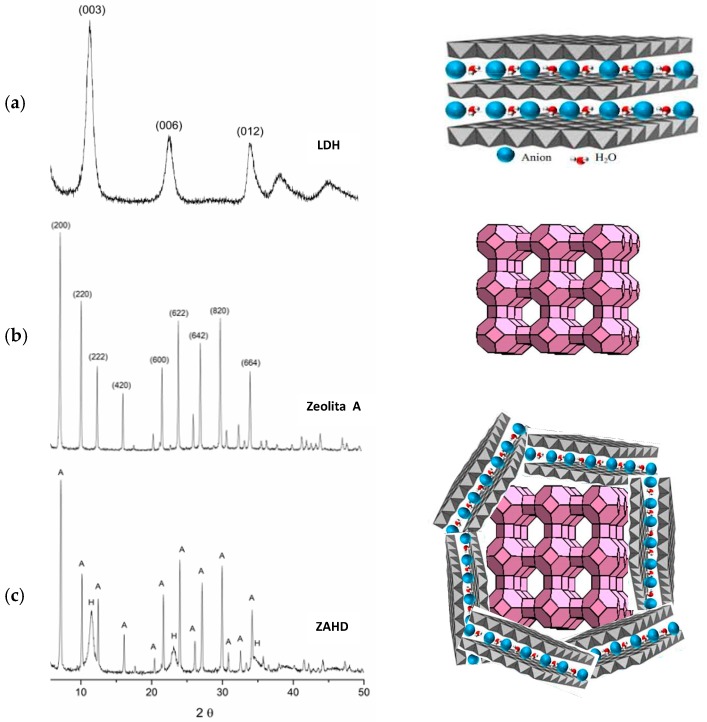
Diffractograms of the structure scheme of the synthesized materials: (**a**) LDH, (**b**) zeolite A, and (**c**) ZAHD composite.

**Figure 2 materials-12-00661-f002:**
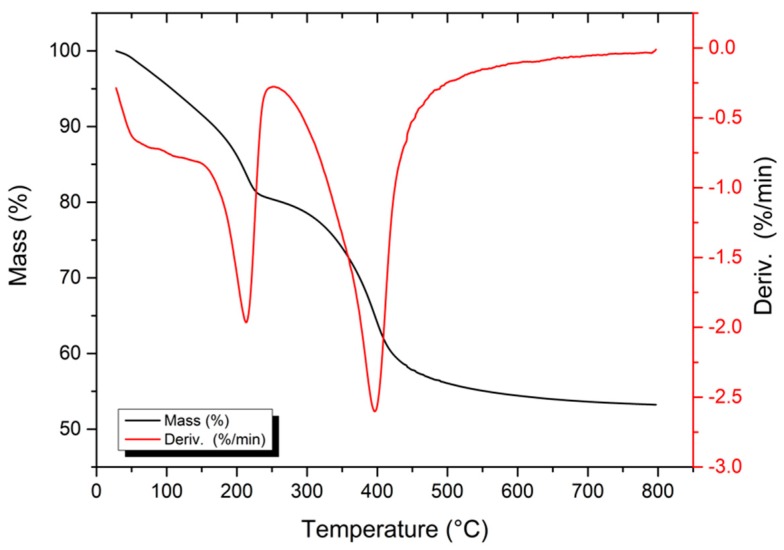
Thermogravimetric and TG/DTG curves of the ZAHD composite.

**Figure 3 materials-12-00661-f003:**
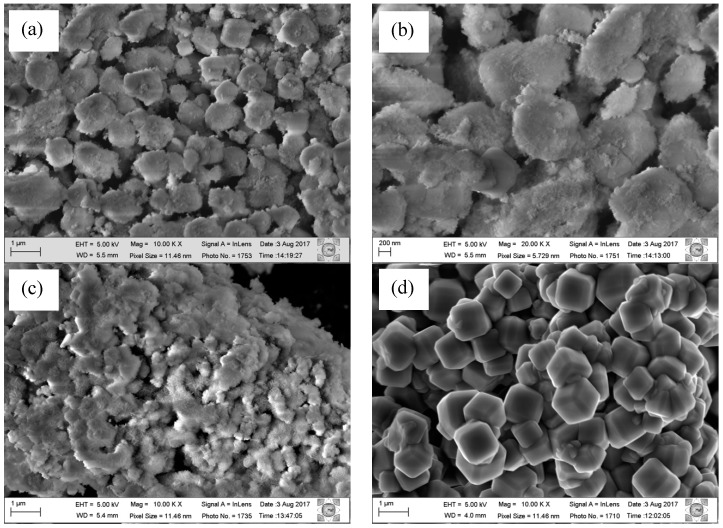
Micrograph of the composite ZAHD (**a**,**b**) material compared to zeolite A (**c**) and LDH (**d**).

**Figure 4 materials-12-00661-f004:**
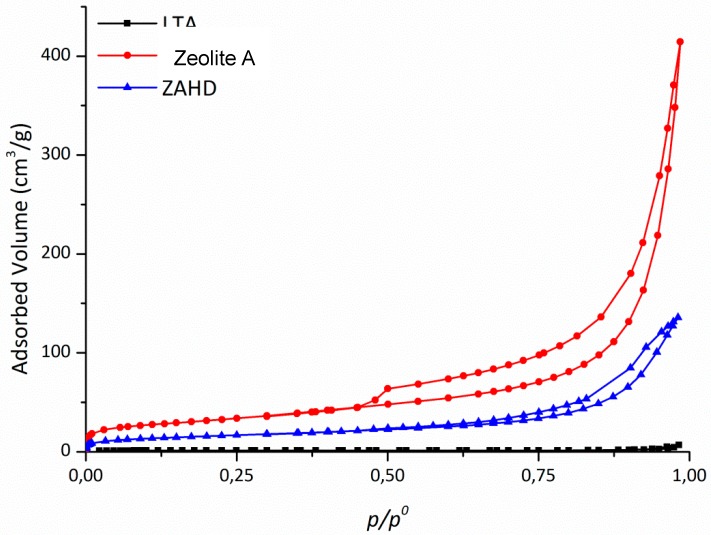
N_2_ isotherms of the synthesized samples: LDH, zeolite A, and ZAHD composite.

**Figure 5 materials-12-00661-f005:**
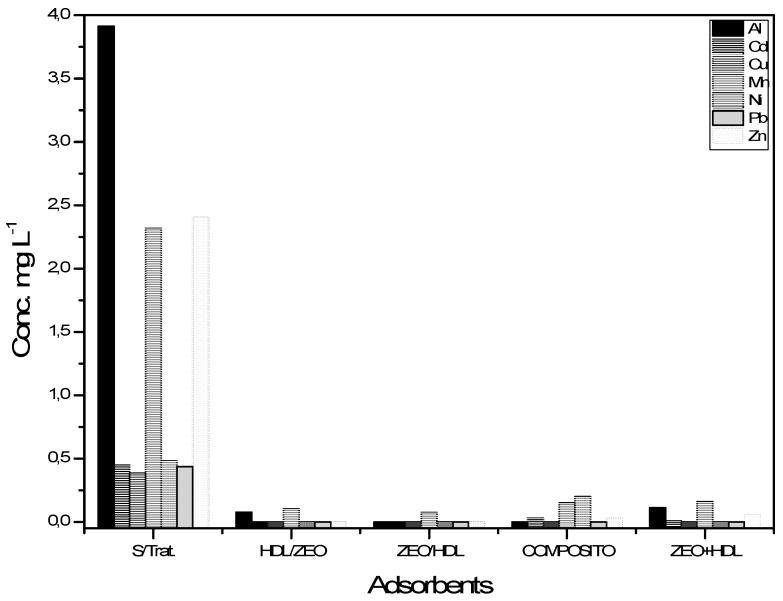
Variations of the concentrations of Al, Cd, Cu, Mn, Ni, Pb and Zn ions in water produced during treatment in the different treatments and analysed by ICP-OES.

**Figure 6 materials-12-00661-f006:**
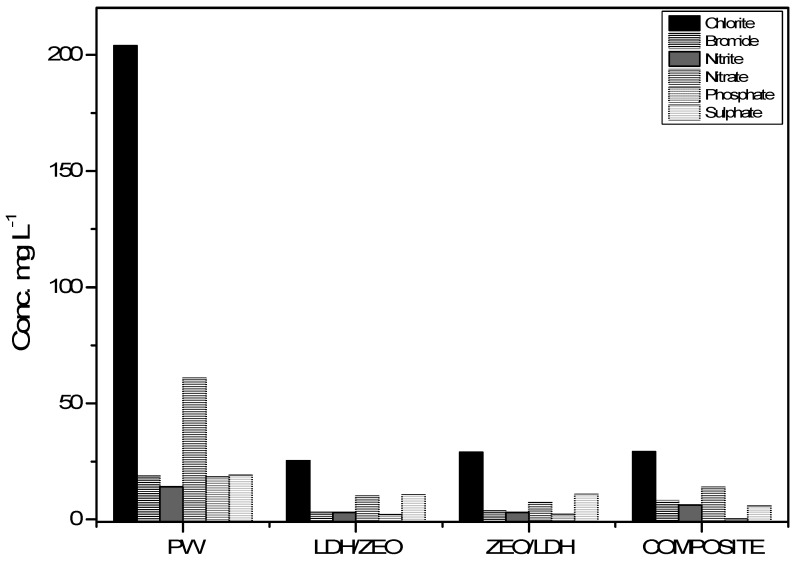
Chloride, bromide, nitrite, nitrate, phosphate and sulphate ion concentrations present in water produced during treatment with the different treatments, as analysed by IC.

**Table 1 materials-12-00661-t001:** FRX results in % for the synthesized materials.

Material	Al_2_O_3_	MgO	SiO_2_	Na_2_O	Si/Al	Mg/Al
Zeolite A	35.50	1.5	45.35	16.9	1.10	0.05
LDH	37.97	58.21	0	3.06	0	1.9
ZAHD	34.71	20.9	33.96	10.0	0.83	0.77

**Table 2 materials-12-00661-t002:** Textural results for the synthesized materials.

Material	S_BET_ (m^2^/g)	VTP (cm^3^/g)/0.98
Zeolite A	5	0.01
LDH	113	0.60
ZAHD	57	0.21

**Table 3 materials-12-00661-t003:** Adsorption results for the synthesized materials.

Parameters	Initial Concentration on PW (mg/L)	ZEO (mg/L)	LDH (mg/L)	LDH/ZEO (mg/L)	ZEO/LDH (mg/L)	COMPOSITE (mg/L)
**Al**	3.913	0	3.930	0.078	0	0
**Cd**	0.448	0	0.446	0	0	0.059
**Cu**	0.385	0	0.384	0	0	0
**Mn**	2.319	0.070	2.322	0.105	0.075	0.740
**Ni**	0.482	0	0.484	0	0	0.131
**Pb**	0.437	0	0.435	0	0	0
**Zn**	2.407	0	0.404	0	0	0.058
Chloride	203.971	200.345	32.965	25.276	28.965	29.138
Bromide	18.656	18.530	3.572	2.982	3.572	8.09
nitrite	13.97	14.02	2.980	2.947	2.98	6.15
nitrate	60.757	60.573	10.239	9.908	7.239	13.731
phosphate	18.304	18.132	2.080	2.019	2.08	0.07
sulphate	19.027	19.123	14.247	10.406	10.647	5.679

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
