# Peer review of "Development of a Zeolite A/LDH Composite for Simultaneous Cation and Anion Removal"

_materials, 2019, doi:10.3390/ma12040661_

Reviewer 1 Report

Manuscript Title:

Development of a zeolite A/LDH composite for simultaneous cation and anion removal

This manuscript studies the new composite material formed by zeolite A and layered double hydroxide for treatment of wastewater from oil industry.

The resulting samples were contrasted through surface area, crystal structure, chemical state and morphology. Finally, the authors evaluated the degradation of cationic orchid X-BL under room temperature and pressure.  The topic is interesting but there are some aspects that the authors should improve.

Introduction:

The introduction should be revised. Since      the paper is focused on the adsorption      process of wastewater from the petroleum industry (PW), in the introduction should be      added some references regarding specifically the issue of adsorption process, underlining the novelty of the paper compering to the authors      prevision paper: In particular, lines 63-69 seem to have no connection      with the previous sentence. Please at this point of introduction explain  why the authors use   zeolite A/LDH composite instead other adsorption materials.

Results:

 Please clarify the      Thermogravimetric analysis. The authors did not report comment about this      type of characterization, what information the authors get?

Please use the same acronyms in all manuscript. For example in Figure      4 who is LTA? Moreover, the LTA isotherm it is not clear, please increase      the quality of Figure 4.

Add a summary table with the experimental results, in this way      would be simpler read and compare the experimental results.

Discussion

What is the adsorption properties of the bare Zeolite A and LDH?      Maybe, you can add this results in the summery table and not only in the      Discussion

Please clarify the sentences:  “This      fact can help to avoid diffusion problems”. Specify why thanks to      composite the authors avoid the diffusion limitations.

Author Response

Thanks a lot for the suggestions, we think that the manuscript is more clear now. We answer (in red) the review with his comments below:

Development of a zeolite A/LDH composite for simultaneous cation and anion removal

This manuscript studies the new composite material formed by zeolite A and layered double hydroxide for treatment of wastewater from oil industry.

The resulting samples were contrasted through surface area, crystal structure, chemical state and morphology. Finally, the authors evaluated the degradation of cationic orchid X-BL under room temperature and pressure.  The topic is interesting but there are some aspects that the authors should improve.

Introduction:

The introduction should be revised. Since      the paper is focused on the adsorption      process of wastewater from the petroleum industry (PW), in the introduction should be      added some references regarding specifically the issue of adsorption process, underlining the novelty of the paper compering to the authors      prevision paper: In particular, lines 63-69 seem to have no connection      with the previous sentence. Please at this point of introduction explain  why the authors use   zeolite A/LDH composite instead other adsorption materials.

We made some modifications on the introduction and we think that now is more clear the novelty and why we use this composite.

Results:

 Please clarify the      Thermogravimetric analysis. The authors did not report comment about this      type of characterization, what information the authors get?

We have this phrase in the text “A thermogravimetric analysis of the composite was performed with the objective of observing the changes in relation to the loss of mass occurring upon heating.” And we put one phrase in discussion “.  So this temperature (400oC) was used for calcination. “.

This characterization is to see the correct temperature for calcination of the material.

Please use the same acronyms in all manuscript. For example in Figure      4 who is LTA? Moreover, the LTA isotherm it is not clear, please increase      the quality of Figure 4.

The reviewer is right, sorry about that. The A zeolite has LTA strucutre, is the same, but we will use only A Zeolite.

About the Figure 4, we put all isotherms toguether to show that Zeolite A has low surface área, because adsorbe low quantity of N2 molecules.

Add a summary table with the experimental results, in this way      would be simpler read and compare the experimental results.

 Ok we add on Table (Table 3) with the experimental results

Discussion

What is the adsorption properties of the bare Zeolite A and LDH?      Maybe, you can add this results in the summery table and not only in the      Discussion

The reviewer is right, we put  on table (table 3)

Please clarify the sentences:  “This      fact can help to avoid diffusion problems”. Specify why thanks to      composite the authors avoid the diffusion limitations.

This is because of the homogeneous size of particles.

Reviewer 2 Report

Reviewed work by Bezerra and coauthors represents a problem of using a zeolite A / LDH composite as sorbents for some metals from wastewaters. The authors promote their research as the first using of new composite formed by zeolite A and LDH to remove some form of cationic and anionic metals from wastewater from the petroleum industry. I have some comments which would improve the presented work.

1. Both in the Abstract and the Introduction the authors characterize wastewater from the petroleum industry. They write that their main component are salts, BTEXs, PAHs, organic acids, phenols and many others. Why with so many pollutants, mainly organic, the authors investigate the sorption of metals, which are just one of many, not the most important pollutions. I would suggest to change the narrative and show that removing metals from these wastewater is actually an important problem to solve.

2. Figure 2. For the full interpretation of the reactions occurring during sample heating, it is also necessary to present the DTG curve.

3. Table 1. The value Mg/Al can not be “0” because the content of MgO is not “0”.

4. Figure 5 and partly Figure 6. The solid-solution ratio used by the authors was so high that the sorption efficiency equals nearly 100% for each sorbent. Such results make it impossible to compare the sorption results and the assessment which sorbent is the most effective. And what are the optimal conditions for effective metal sorption.

5. Lines 280-286. This part of the manuscript is completely incomprehensible to me. In the methodological description (Section 2.2), the authors do not describe conditions presented in this place (lines 280-286) and therefore it is not known what the mass of the samples and reaction times given by the authors mean. Please describe it more clearly. In addition, the data presented are not consistent with the data presented in Figure 5 and Figure 6.

6. The authors do not mention anything about sorption mechanisms. This is a basic part of the description of the use of new sorbents or new composites in sorption experiments.

Author Response

Thanks for the suggestions, we think that the manuscript is more clear now. Following we answer the review (in red):

Reviewed work by Bezerra and coauthors represents a problem of using a zeolite A / LDH composite as sorbents for some metals from wastewaters. The authors promote their research as the first using of new composite formed by zeolite A and LDH to remove some form of cationic and anionic metals from wastewater from the petroleum industry. I have some comments which would improve the presented work.

1.     Both in the Abstract and the Introduction the authors characterize wastewater from the petroleum industry. They write that their main component are salts, BTEXs, PAHs, organic acids, phenols and many others. Why with so many pollutants, mainly organic, the authors investigate the sorption of metals, which are just one of many, not the most important pollutions. I would suggest to change the narrative and show that removing metals from these wastewater is actually an important problem to solve.

We agree with the reviewer, we only intent to show the complex matrix that PW is.  We change adding and remove some phrases to be more clear that we want to remove metals, cátions and anions.

2. Figure 2. For the full interpretation of the reactions occurring during sample heating, it is also necessary to present the DTG curve. In figure 2 is present the loss weight curve (TG analysis) and the derivative curve that is DTG curve. So we can take the temperature range that the lost weight event occurs.

3. Table 1. The value Mg/Al can not be “0” because the content of MgO is not “0”.

The reviewer is right, sorry about that. The value is 0.05. We put this on Table 1.

4. Figure 5 and partly Figure 6. The solid-solution ratio used by the authors was so high that the sorption efficiency equals nearly 100% for each sorbent. Such results make it impossible to compare the sorption results and the assessment which sorbent is the most effective. And what are the optimal conditions for effective metal sorption.

The reviewer is right, in previous studies we made kinetic studies and diferente solid-solution ratio, and we can observe diferente adsorptions capacities. And we choose this condicions because they are the best, with maximun adsorption. So when we compare these materials, they don’t have so much changes, because they are the same materials but in diferente processes, or in secuential or in composite form. In this way, we can say that it is importante to use both materials Zeolite A and LDH for cátions and anions remove. They works in secuential form and in composite form. In composite form will be preferer because that is only one material and that can be reuse and regenerated more simplier than reuse and regenerate two materials in  a secuential form. We want to show that the composite has the same removal capacity than separeted materials in sequencial form and in less time.

5. Lines 280-286. This part of the manuscript is completely incomprehensible to me. In the methodological description (Section 2.2), the authors do not describe conditions presented in this place (lines 280-286) and therefore it is not known what the mass of the samples and reaction times given by the authors mean. Please describe it more clearly. In addition, the data presented are not consistent with the data presented in Figure 5 and Figure 6.

In section 2.2 we sad that: The treatment of the water produced from oil (from the RN CE Operation Unit) was performed using 50 mL of the solution in contact with a mass of 0.30 g of each material. The samples were placed in 150 mL Erlenmeyer flasks and subjected to a stirring orbital shaker table operating at 200 rpm for periods equal to 4 h.  

So that means that we use 0,3 g and 4h for each treatment, when we use only zeolite A and  only LDH. When we use the sequential method, we use 0,3g of zeolite A and then 0,3g of LDH, for 4 each, total was 8h. When we use de composite for comparison we used 0,6 g of composite and 4h.

Have some mistatek on the text it is not 6g and 0,6g.

6. The authors do not mention anything about sorption mechanisms. This is a basic part of the description of the use of new sorbents or new composites in sorption experiments.

The reviewer is right. We add a phrase explaining that. In these kind of materials the mechanism is cation or anion exchanges. So it is important que  Si/Al ratio on zeolites, because with more Al quantities, more cation exchange capacity it will have. In the case of LDH, the major contente of Al will give more positive charge to layer and more anion exchange capacity. Both materials have also –OH that can also retain cátions by eletrostatic interacions.

Round  2

Reviewer 1 Report

Accept in present form

Author Response

Thankyou.

Reviewer 2 Report

No comments and suggestions

Author Response

Thankyou